# Three body photodissociation of the water molecule and its implications for prebiotic oxygen production

Yao Chang [1,8], Yong Yu[1,8], Feng An[2], Zijie Luo[1,3], Donghui Quan[4], Xia Zhang [5], Xixi Hu [2✉], Qinming Li[1], Jiayue Yang[1], Zhichao Chen[1], Li Che[3], Weiqing Zhang[1], Guorong Wu [1], Daiqian Xie [2], Michael N. R. Ashfold [6], Kaijun Yuan [1✉] & Xueming Yang [1,7✉]

The provenance of oxygen on the Earth and other planets in the Solar System is a fundamental issue. It has been widely accepted that the only prebiotic pathway to produce oxygen in the Earth's primitive atmosphere was via vacuum ultraviolet (VUV) photodissociation of $CO_2$ and subsequent two O atom recombination. Here, we provide experimental evidence of three-body dissociation (TBD) of $H_2O$ to produce O atoms in both [1]D and [3]P states upon VUV excitation using a tunable VUV free electron laser. Experimental results show that the TBD is the dominant pathway in the VUV $H_2O$ photochemistry at wavelengths between 90 and 107.4 nm. The relative abundance of water in the interstellar space with its exposure to the intense VUV radiation suggests that the TBD of $H_2O$ and subsequent O atom recombination should be an important prebiotic $O_2$-production, which may need to be incorporated into interstellar photochemical models.

[1] State Key Laboratory of Molecular Reaction Dynamics and Dalian Coherent Light Source, Dalian Institute of Chemical Physics, Chinese Academy of Sciences, Dalian, China. [2] Key Laboratory of Mesoscopic Chemistry, School of Chemistry and Chemical Engineering, Institute of Theoretical and Computational Chemistry, Nanjing University, Nanjing, China. [3] Department of Physics, School of Science, Dalian Maritime University, Dalian, Liaoning, China. [4] Eastern Kentucky University, Richmond, KY, USA. [5] Xinjiang Astronomical Observatory, Chinese Academy of Sciences, Urumqi, China. [6] School of Chemistry, University of Bristol, Bristol, UK. [7] Department of Chemistry, College of Science, Southern University of Science and Technology, Shenzhen, China. [8] These authors contributed equally: Yao Chang, Yong Yu. ✉email: xxhu@nju.edu.cn; kjyuan@dicp.ac.cn; xmyang@dicp.ac.cn

Oxygen is the third most abundant element in the Universe, but its molecular form ($O_2$) is very rare. Besides on the Earth, molecular oxygen has only been detected in two interstellar clouds[1,2], in the moons of Jupiter[3] and Saturn[4], and on Mars[5]. Geologically based arguments suggested that Earth's original atmosphere had no oxygen and was composed mostly of $H_2O$, $CO_2$, and $N_2$, with only small amounts of CO and $H_2$[6]. Therefore, how oxygen is produced in the primitive atmosphere is a fundamentally important issue in the evolution of the early primitive atmosphere. Before the emergence of the oxygen-rich atmosphere due to the "great oxidation event"[7,8], about 2.33 billion years ago, which allowed the Earth to evolve into a living planet, a small amount of oxygen was already present, and this was previously attributed to an abiotic formation mechanism involving photodissociation of $CO_2$ by vacuum ultraviolet (VUV) light, followed by three-body recombination processes[9]:

$$CO_2 + h\nu \rightarrow CO + O \quad (1)$$

$$O + O + M \rightarrow O_2 + M \quad (2)$$

where M is a third body to carry off the excess energy in the reaction process. Direct $O_2$ production pathways via VUV photodissociation of $CO_2$[10] and dissociative electron attachment to $CO_2$[11] have recently been identified. These findings provide new insights into the sources of $O_2$ in Earth's early atmosphere.

In contrast, photodissociation of $H_2O$, one of the dominant oxygen carriers[12], has long been assumed to proceed mainly to produce hydroxyl (OH) and hydrogen (H) atom primary products, and contribute limitedly to the $O_2$ production[9]. Recently, abundant molecular $O_2$ in the coma of comet 67P/Churyumov–Gerasimenko, which is dominated by $H_2O$, CO, and $CO_2$, has been detected[13]. Interestingly, a strong correlation between $O_2$ and $H_2O$ has been identified, indicating the $O_2$ formation is linked to $H_2O$ in the comet. One plausible explanation for the strong $O_2$–$H_2O$ correlation would be that the $O_2$ is produced by radiolysis or photolysis of water, or single collisions of energetic $H_2O^+$ with surfaces[14]. However, the existing photochemical reaction mechanisms underestimate the $O_2$ abundance[13]. Thus, the detailed $O_2$ production mechanism in the coma of comets is still unclear.

The photodissociation of water has been the subject of many experimental studies, which have revealed fascinating dynamics arising from strongly coupled electronic states with strikingly different potential energy surfaces (PESs)[15,16]. Excitation to the first excited singlet ($\tilde{A}^1B_1$) state of $H_2O$ at wavelengths $\lambda$ ~160 nm results in direct O−H bond fission yielding an H atom plus a ground state hydroxyl radical, $OH(X^2\Pi)$, with little internal excitation[17–19]. The absorption cross-section to the second excited singlet ($\tilde{B}^1A_1$) state is maximal at $\lambda$ ~128 nm. Excitation at these wavelengths results in a (minor) direct dissociation channel to electronically excited $OH(A^2\Sigma^+) + H$ products. The major dissociation process yields ground state $OH(X) + H$ products following non-adiabatic transitions at conical intersections between the $\tilde{B}$ and $\tilde{X}$ state PESs at linear H–O–H and H–H–O geometries[20–23].

Additional fragmentation pathways, named three-body dissociation (TBD), become accessible energetically at shorter photolysis wavelengths, e.g.:

$$H_2O + h\nu \rightarrow O(^3P) + H + H \rightarrow (9.513 \text{ eV}) \quad (3)$$

$$\rightarrow O(^1D) + H + H \rightarrow (11.480 \text{ eV}) \quad (4)$$

where the threshold energies ($E_{th}$) for these fragmentation channels are given in parentheses (on the basis of thermodynamic calculations with the data available from the thermochemical network) (https://atct.anl.gov). Fragmentation channel (3) has been previously detected with small quantum yields[24,25] following

photoexcitation of $H_2O$ at the Lyman-α wavelength ($\lambda = 121.57$ nm). Because of the lack of intense tunable VUV laser sources, the quantitative assessment of the importance of the $H_2O$ TBD processes in the VUV region and its role in the $O_2$ formation in the interstellar space has not been possible. Recent development of the intense VUV free electron laser, at the Dalian Coherent Light Source (DCLS), has provided an exciting tool for experimental studies of molecular photochemistry throughout the entire VUV region[26].

Herein we report the studies of the $H_2O$ photochemistry in the VUV region using the DCLS and the H-atom Rydberg tagging time-of-flight (HRTOF) technique. These experiments allow quantitative determination of the relative importance of the binary dissociation and the TBD processes following photoexcitation of $H_2O$ in the 90–110 nm region. The present results show conclusively that the $H_2O$ TBD process is an important pathway to form oxygen in the interstellar space.

## Results and discussion

The $H_2O$ sample was generated in a supersonic beam, with a rotational temperature estimated to be about 10 K. The $H_2O$ molecules were photoexcited to different Rydberg states[27] (see Supplementary Fig. 2 and Supplementary Note 2). The dissociated H atom fragments were then detected using the HRTOF technique (see Supplementary Fig. 1 and Supplementary Note 1). TOF spectra of the H atoms resulting from $H_2O$ photodissociation at $\lambda = 107.4$ nm have been recorded, with the detection axis aligned parallel and perpendicular to the polarization vector of the VUV-FEL radiation. Knowing both the distance traveled by the H atom from the photodissociation area to the detector and its mass, the TOF spectra can be converted into the distributions of total kinetic energy release (TKER), which is derived using the following equation, $E_{KE} = \frac{1}{2}m_H(1 + \frac{m_H}{m_{OH}})(\frac{d}{t})^2$, where $d$ is the flying path length of H atom ($d \approx 28$ cm) from the photodissociation region to the detector, $t$ is the measured time of flight[28]. Using the TKER distributions obtained in the parallel and perpendicular directions, we can construct a 3-dimensional (3D) flux diagram of the H atom fragments. Figure 1 shows the 3D product flux diagrams in two regions of the kinetic energy with rich structures. The tall feature at low kinetic energy (Fig. 1A) shows a large product angular anisotropy, whereas the product anisotropy in the higher kinetic energy region (Fig. 1B) is rather small.

For detailed analysis and feature assignment, the TKER distributions in parallel and perpendicular directions at $\lambda = 107.4$ nm are plotted in Fig. 1C, while the product TKER distribution from photodissociation of $H_2O$ at the magic angle (with detection angle of 54.7° relative to the polarization direction) is shown in Fig. 1D. These distributions show both sharp and broad features. Using the energy conservation relationship appropriate for a binary photodissociation process:

$$E_{h\nu} + E_{int}(H_2O) = D_0(H - OH) + E_{KE}(H + OH) + E_{int}(OH)$$

$$(5)$$

where $E_{int}(H_2O)$ and $E_{int}(OH)$ are the internal energies of $H_2O$ and OH, respectively, $E_{KE}$ is the product total kinetic energy and $D_0(H-OH)$ is the dissociation energy of $H_2O$[29]. We can assign all of the sharp structures to specific ro-vibrational levels of the OH product in the X and A states formed via the binary dissociation channel, H + OH (X or A, $v$, $N$). In addition to these sharp structures, the TKER spectra show two broad features: one with $E_{KE} \leq 600$ cm$^{-1}$ that has a large angular anisotropy, and an underlying feature that spans the range of $600 \leq E_{KE} \leq 16,000$ cm$^{-1}$ which displays a much smaller angular anisotropy. These broad features are obviously not from the binary dissociation channel, H + OH. Based on energy conservation, the maximum kinetic energy for the H atom product

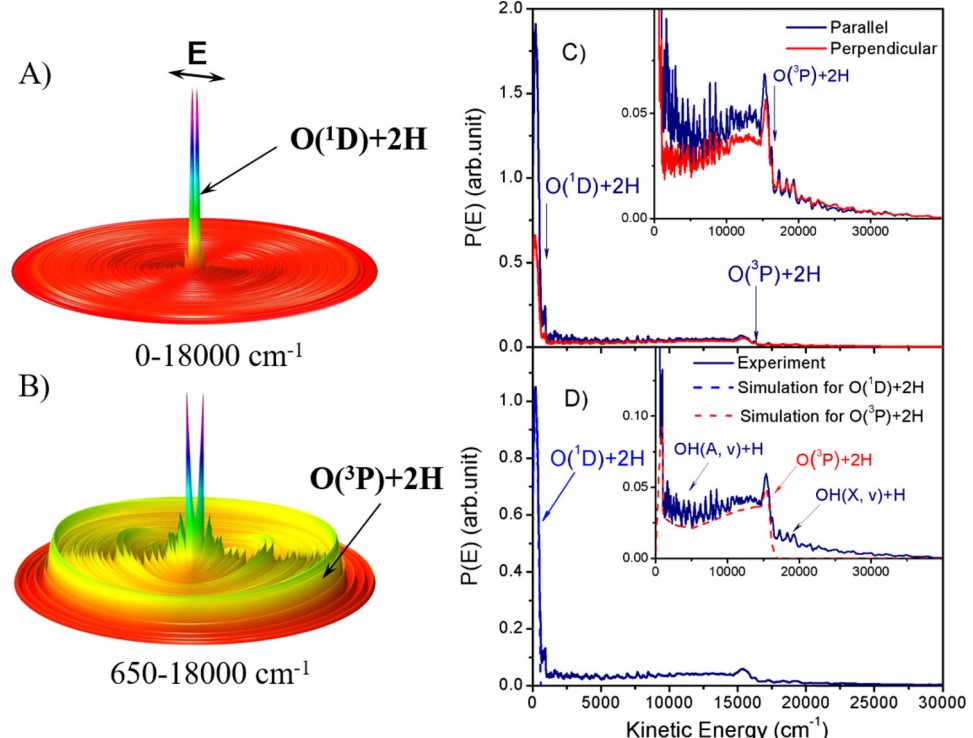

**Fig. 1 The kinetic energy release spectra from H₂O photodissociation. A** The 3D product contour diagram from the photodissociation of $H_2O$ at 107.4 nm for the total kinetic energy release (TKER) between 0 and 18,000 cm⁻¹; **B** the 3D product contour diagram for the TKER between 650 and 18,000 cm⁻¹. **C** The product TKER distributions at $\lambda = 107.4$ nm with the detection axis parallel and perpendicular to the polarization vector of the VUV-FEL radiation. The energetic limits of the two TBD channels ($E_{KEmax}$ ~580 and ~16,448 cm⁻¹) are marked, and the inset displays the same spectra on an expanded vertical scale. **D** The product TKER distribution at $\lambda = 107.4$ nm with the detection axis at 54.7° (magic angle) to the polarization direction, along with the simulated TKER distributions for the O($^1$D) + 2H and O($^3$P) + 2H TBD channels. Source data are provided as a Source Data file.

from the TBD channels (3) and (4) at 107.4 nm are 16,448 and 580 cm⁻¹, respectively. These limits of the two TBD channels match well with the upper limits of the two broad features in the distributions (Fig. 1). Thus, the intense broad feature at $E_{KE} < 600$ cm⁻¹ is assigned to the O($^1$D) + 2H channel, while the broad underlying signal extending to $E_{KE}$ ~16,000 cm⁻¹ is attributed to the O($^3$P) + 2H products.

Given the above analysis, the product TKER distribution in Fig. 1D can be divided into three components: O($^1$D) + 2H, O($^3$P) + 2H, and OH + H. The first two components are broad features, while the third comprises sharp structures attributable to ro-vibrational levels of the OH(A) and OH(X) products. From Fig. 1D, the O($^1$D) + 2H channel obviously has the highest intensity in the kinetic energy <600 cm⁻¹, yet it has never been reported previously in H₂O photodissociation. The O($^3$P) + 2H channel clearly makes the major contribution in the higher kinetic energy region. This indicates that the TBD is likely the dominant process following VUV photoexcitation of H₂O at 107.4 nm.

Branching ratios for the O($^1$D) + 2H, O($^3$P) + 2H, and OH + H fragmentation channels have been estimated by simulating the TKER distribution (Fig. 1D) using the three components shown in Fig. 2. Integrating the areas under the respective distributions returns relative H atom yields for the three channels. Recognizing that two H atoms are formed in each TBD process, the relative H atom yields are used to determine the branching ratio, e.g. 67% at 107.4 nm for the TBD channels (Table 1).

Photodissociation of H₂O has also been investigated at eight more VUV wavelengths between 92 and 109 nm, and a similar data analysis procedure is applied at these photolysis wavelengths (Supplementary Fig. 3 and Supplementary Note 3). The branching

ratios determined for the binary and TBD channels at each wavelength are listed in Table 1. At 109.0 nm, only one TBD channel (O ($^3$P) + 2H) exists because the O($^1$D) + 2H channel is energetically not accessible. The results mean that oxygen atoms ($^1$D and $^3$P), not OH radicals, are the major oxygen-containing products from H₂O photolysis at $\lambda < 107.4$ nm, in striking contrast to the dominant binary photofragmentation (i.e. H + OH) behavior displayed by H₂O photochemistry at longer VUV wavelengths[15].

The dissociation dynamics of the two TBD channels are also quite interesting. Since the two H atoms in the water molecule are equivalent, if they dissociate simultaneously it should yield a narrow H atom kinetic energy distribution, peaking at an $E_{EK}$ value close to half of the available energy (Supplementary Figs. 5 and 6 and Supplementary Note 5). However, the observed distributions are much broader than the narrow distributions for a simultaneous concerted process, implying that both TBD processes are due to mostly a sequential dissociation mechanism. Possible dissociation routes for the two channels are illustrated in Fig. 3 (and more details in Supplementary Fig. 7 and Supplementary Note 6). The water molecule undergoes efficient nonadiabatic coupling from the initial excited $nd$ Rydberg states to the $\widetilde{D}$ state. Path 1 illustrates the possible direct dissociation route from the $\widetilde{D}$ state to form O($^1$D) + 2H products. The overall anisotropy parameter of this channel is about 0.8, according with a fast and direct dissociation process. Paths 2 and 3 illustrate plausible routes for the more complicated O($^3$P) + 2H dissociation paths: from the $\widetilde{D}$ state to the $\widetilde{B}$ state and then to the ground state via the two CIs between the $\widetilde{B}$ and $\widetilde{X}$ state PESs, following the initial internal conversion from the excited $nd$ Rydberg states to the $\widetilde{D}$ state. The averaged anisotropy parameter of this channel

is considerably smaller (~0.2). This also accords with the dissociation undergoing several internal conversions, and with the more scrambling that leads to small product anisotropy.

The conclusion that the TBD is the dominant decay process following excitation of $H_2O$ at these VUV wavelengths could have profound implications for our understanding of the source of oxygen production. For quantitative assessment, we have calculated the fragment-dependent photodissociation rate of $H_2O$ by using: $J_{H2O} = \int \Phi_\lambda \Gamma \sigma_\lambda d\lambda$, where $\Phi_\lambda$ is the solar photon flux, $\Gamma$ is the fragment quantum yield, and $\sigma_\lambda$ is the photodissociation cross-section[30]. Figure 4 collects together the wavelength dependences of the solar photon flux in the early period[31], the total photoabsorption cross-sections of the parent $H_2O$ molecule in the VUV region (90–200 nm)[32] and the production yields of O

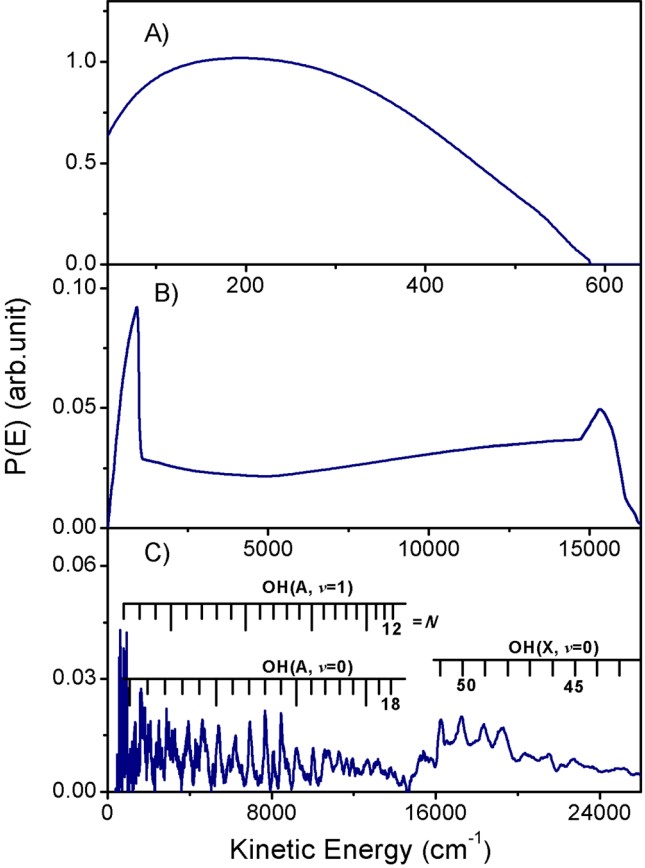

**Fig. 2 The simulated kinetic energy release spectra from $H_2O$ photodissociation.** TKER distributions determined from simulating the (**A**) $O(^1D) + 2H$, (**B**) $O(^3P) + 2H$, and (**C**) $OH + H$ product yields from photodissociation of $H_2O$ at $\lambda = 107.4$ nm. The sharp features in the H + OH product yield have been assigned to population of ro-vibrational levels of the X and A states of the OH radical. Source data are provided as a Source Data file.

atoms at studied photolysis wavelengths. Convoluting the solar photon flux, the photoabsorption cross-sections, and the production yields implies that ~21% of the photoexcitation events of $H_2O$ will result in O atoms. Considering the water abundance in widely interstellar circumstances, like in interstellar clouds[2,3] and in the moons of planets in the Solar System (e.g., the comets 67P[13,33]), oxygen production from the water photolysis should be an important process. The following recombination of oxygen atoms will produce molecular oxygen.

In addition, it is well known that the water photolysis has nothing to do with oxygen production in the Earth's atmosphere under equilibrium conditions due to VUV photon screening by the thick atmosphere[10,34]. However, in the earliest period of Earth, i.e., the period approaching to clement conditions on the earliest Earth followed by the current Earth–Moon system formed, the surface of Earth remained quite hot (>1000 K)[34], all of the water on the Earth was vaporized to the atmosphere and part of water clouds (emitted from volcanos or delivered by carbonaceous chondrite meteorites[6]) populated at the top of the atmosphere could absorb the VUV photons and dissociate. Given $[H_2O]$ is ten times abundant than $[CO_2]$ in the atmosphere during this early, chaotic period of Earth[6] (see Supplementary Fig. 4 and Supplementary Note 4), the O production rate from $H_2O$ VUV photochemistry could be three times larger than that of $CO_2$ in the same VUV wavelength region, via TBD processes: $N_{H2O}(O)/N_{CO2}(O) = (J_{H2O}(O) \times [H_2O])/(J_{CO2}(O) \times [CO_2]) = \sim 3$, where $J_{H2O}(O) = \sim 5.2 \times 10^{-5}\,s^{-1}$ (Fig. 4), $J_{CO2}(O) = \sim 1.8 \times 10^{-4}\,s^{-1}$ (Supplementary Fig. 8 and Supplementary Note 7), $[H_2O]$ and $[CO_2]$ are the densities of $H_2O$ and $CO_2$, respectively. Since the molecular oxygen generation process should be the same in the three-body recombination process (Eq. (2)), this analysis implies that the $H_2O$ photochemistry might be an important prebiotic source of $O_2$ in Earth's early atmosphere.

From the experimental results, it seems that more than one-third of O atoms produced from $H_2O$ TBD process populate in the metastable $^1D$ state. The generation of $O(^1D)$ atoms from photodissociation of $H_2O$ in a significant amount is also very interesting because the metastable $O(^1D)$ atom is highly reactive[35]. It can react with almost all the gases emitted into the atmosphere. For instance, the reaction of $O(^1D)$ with methane could be a significant source of formaldehyde in the earth's primitive atmosphere[36,37]. Thus, the production of $O(^1D)$ atoms from the exposure of water to VUV radiation, and the subsequent reactions of these atoms, could have been important drivers in the evolution of the earliest atmosphere.

In the existing interstellar photochemical model, reactions (1) and (2) are the major pathways to produce prebiotic $O_2$. In this work, we propose an alternative prebiotic $O_2$ pathway: atomic oxygen production from the TBD of water, followed by oxygen recombination reactions. Recent International Ultraviolet Explorer satellite observation of pre-main-sequence stars suggested that the nascent Sun has emitted more than ten times VUV radiation than it does today[38]. This implies that oxygen formation by VUV photoinduced TBD of $H_2O$ is likely an

**Table 1 The branching ratios for the binary and TBD channels following the $H_2O$ photodissociation at different VUV wavelengths (nm).**

| Dissociation channel | Photolysis wavelength (nm) | | | | | | | | |
|---|---|---|---|---|---|---|---|---|---|
| | 109.0 | 107.4 | 106.7 | 105.7 | 101.3 | 98.1 | 96.2 | 94.5 | 92.0 |
| TBD ($O(^1D/^3P) + 2H$) | 0.35 | 0.67 | 0.76 | 0.62 | 0.77 | 0.72 | 0.79 | 0.86 | 0.86 |
| Binary dissociation: (H + OH) | 0.65 | 0.33 | 0.24 | 0.38 | 0.23 | 0.28 | 0.21 | 0.14 | 0.14 |

The maximum uncertainty on the branching ratios is ±10%.

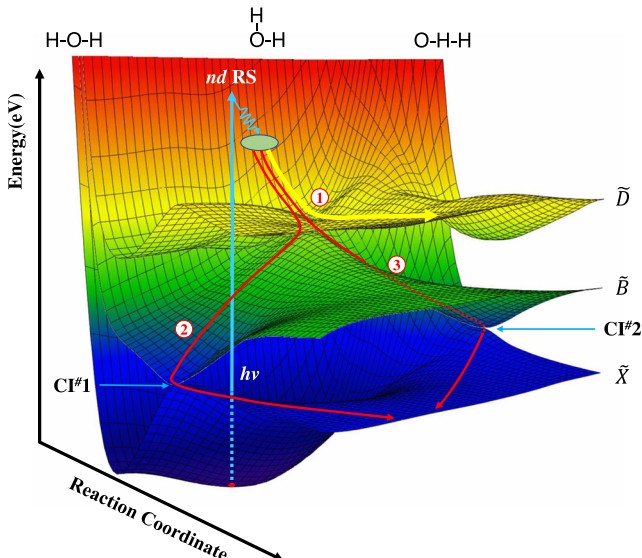

**Fig. 3 Illustration of the TBD mechanisms of H₂O upon VUV excitation.**
Photoexcitation populates the *nd* Rydberg state (*nd* RS) which undergo efficient non-adiabatic coupling to the $\widetilde{D}$ state. The light blue arrow displays the photoexcitation process and the gray oval represents the crossing region between the *nd* RS and the $\widetilde{D}$ state. Path 1 illustrates the possible direct dissociation route from the $\widetilde{D}$ state to form $O(^1D) + 2H$ products (the yellow curve). Paths 2 and 3 illustrate plausible routes for the more complicated $O(^3P) + 2H$ dissociation paths: from the $\widetilde{D}$ state to the $\widetilde{B}$ state and then to the ground state via the two conical intersections (CI #1 or CI #2), following the initial internal conversion from the excited *nd* RS to the $\widetilde{D}$ state (the red curves). This picture is consistent with the observed angular anisotropy for the two channels: the $O(^1D) + 2H$ channel is a more direct dissociation process with larger angular anisotropy, while the $O(^3P) + 2H$ channel is a more complicated dissociation process with smaller angular anisotropy. The PES contour color from blue to red represents the potential energy from 0 to 12 eV.

important process in the coma of comets, in the interstellar clouds and even in Earth's primitive atmosphere, and thus needs to be incorporated into interstellar photochemical models. Furthermore, the TBD of H₂O may well be important for the oxygen evolution in the atmospheres of all water-rich terrestrial planets[39].

## Methods

**Vacuum ultraviolet free electron laser (VUV-FEL) radiation.** The experiments employ a recently constructed apparatus for molecular photochemistry, which is centered on the VUV-FEL beam line at the DCLS[26]. The VUV-FEL facility runs in the high gain harmonic generation mode, in which the seed laser is injected to interact with the electron beam in the modulator (Supplementary Fig. 1). The seeding pulse, in the wavelength range ($\lambda_{seed}$) 240–360 nm, can be generated from a picosecond Ti:sapphire laser pulse. The electron beam is generated from a photocathode RF gun, and accelerated to the beam energy of ~300 MeV by seven S-band accelerator structures, with a bunch charge of 500 pC. The micro-bunched beam is then sent through the radiator, which is tuned to the 2nd/3rd/4th harmonic of the seed wavelength, and coherent FEL radiation with wavelength $\lambda_{seed}/2$, $\lambda_{seed}/3$, or $\lambda_{seed}/4$ is emitted. Optimization of the linear accelerator yields a high quality electron beam with emittance of ~1.5 mm mrad, energy spread of ~1‰, and pulse duration of ~1.5 ps. In this work, the VUV-FEL operates at 10 Hz, and the maximum pulse energy is >100 μJ/pulse. The output wavelength is continuously tunable in the range 50–150 nm and the typical spectral bandwidth of the VUV-FEL output is 30–50 cm⁻¹.

The high-*n* HRTOF technique used in this work was pioneered by Welge et al.[40]. The key point of this technique is the 1 + 1′ (VUV + UV) excitation of the H atom. The first step involves VUV laser excitation of the H atom from its *n* = 1 ground state to the *n* = 2 state by absorbing one $\lambda$ = 121.57 nm photon. In the second step, the H (*n* = 2) atom is excited with a UV ($\lambda$ ~365 nm) photon to a high-*n* (*n* = 30–80) Rydberg state. Charged species formed in the interaction

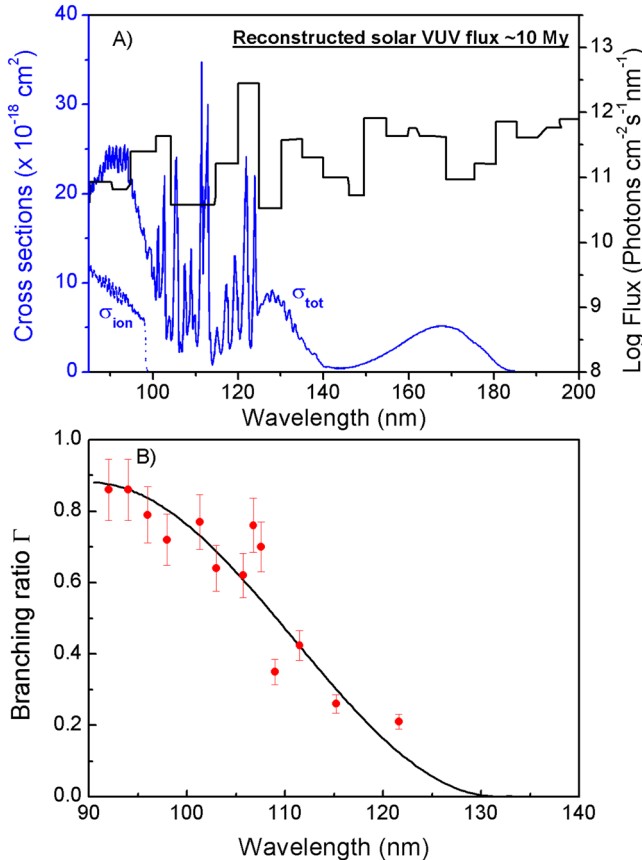

**Fig. 4 The wavelength dependences of O atom quantum yields.** Plot showing (**A**) the wavelength dependences of the reconstructed VUV solar flux (90–200 nm, the black lines) at ~10 My (10 My = 1×10⁷ years, reconstructed from ref. [31]. The VUV solar flux at modern period or the interstellar radiation field (ISRF)[41] also can be used, which may modify the yield of O-production a little, but the final conclusion holds), the total absorption ($\sigma_{tot}$, the solid blue curve)[32] and photoionization ($\sigma_{ion}$, the dotted blue curve) cross-sections[42] of H₂O, and (**B**) the quantum yield for forming O-atom photoproducts ($O(^3P/^1D) + 2H$), $\Gamma$, determined in the present work (the red dot). It is noted that the predissociation rate of H₂O is sufficiently fast that the fluorescence quantum yield must be negligible, so the total photodissociation cross-section will be almost the same as the photoabsorption cross-section. The polynomial function through the latter data is used to derive the reported overall O product quantum yield (the black curve in **B**). The quantum yields at $\lambda$ = 111.5, 115.2, 121.57 nm are obtained from refs. [15, 24, 26]. The error bars represent the standard deviation (1$\sigma$) of three times measurements. Source data are provided as a Source Data file.

region are extracted from the TOF axis by a small electric field (~20 V/cm) placed across this region. Rydberg tagged neutral H atoms fly a known distance (*d* ≈280 mm) from the interaction region to a rotatable microchannel plate (MCP) Z-stack detector located close behind a grounded fine metal grid. After passing through the grid, the Rydberg atoms are immediately field-ionized by the electric field (~2000 V/cm) applied between the grid and the front plate of the Z-stack MCP detector. The signal detected by the MCP is amplified by a fast preamplifier and counted by a multichannel scaler.

## Data availability

All other data supporting this study are available from the authors upon request. Source data are provided with this paper.

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

## Acknowledgements

The experimental work was supported by the National Natural Science Foundation of China (NSFC Center for Chemical Dynamics (Grant No. 21688102)), the National Natural Science Foundation of China (Grant Nos. 21873099, 21922306), the Strategic Priority Research Program of the Chinese Academy of Sciences (Grant No. XDB17000000), the Key Technology Team of the Chinese Academy of Sciences (Grant No. GJJSTD20190002), the Liaoning Revitalization Talents Program (XLYC1907154), and the international partnership program of Chinese Academy of Sciences (No. 121421KYSB20170012). The theoretical work was supported by the National Natural Science Foundation of China (Grant Nos. 21733006, 22073042, and U1932147). M.N.R.A. is grateful for funding from the Engineering and Physical Sciences Research Council (EPSRC, EP/L005913) and to the NFSC Center for Chemical Dynamics for the award of a Visiting Fellowship.

## Author contributions

K.Y. and X.Y. designed the experiments. Y.C., Y.Y., Z.L., Q.L., J.Y., and Z.C. performed the experiments. X.Z., D.Q., K.Y., M.N.R.A., L.C., W.Z., G.W., and X.Y. discussed the experimental results. F.A., X.H., and D.Q. performed the theoretical calculations. K.Y., M.N.R.A., and X.Y. prepared the manuscript.

## Competing interests

The authors declare no competing interests.
