## [Peer Review File · Nature Communications]

REVIEWER COMMENTS

Reviewer #1 (Remarks to the Author):

The manuscript submitted by Chang et al. represents an outstanding contribution in an important problem regarding the provenance of molecular oxygen on Earth and other planets by a detailed study of the three-body photodissociation of water by using tunable VUV free electron laser radiation in combination with the high resolution hydrogen-atom Rydberg tagging time-of-flight technique. These unique experimental capabilities have allowed the determination of quantum yields for the two possible three-body photodissociation channels at different excitation wavelengths in the range 92-109 nm, which have permitted to evaluate the role of VUV three-body photodissociation of water in the production of prebiotic molecular oxygen in Earth's primitive atmosphere. These results will be very important for the interstellar photochemical models. The paper is very good written, the results are clearly presented with the help of the supplementary material and the conclusions are supported by the data. I recommend publication of this work as it is.

Reviewer #2 (Remarks to the Author):

The manuscript reports a new pathway for oxygen production by three body dissociation of H₂O molecules with VUV radiation utilizing the newly developed free electron laser (FEL) at the Dalian Coherent Light Source (DCLS). This is the first comprehensive quantitative observation of TBD of water molecules. I believe this study is greatly important for the better understanding of different oxygen sources in the atmosphere therefore, beneficial for future investigations. Since O atom is a precursor for many atmospheric molecules this will also lay a ground work for studying new photochemical processes. Therefore, I strongly recommend the article for the publication of Nature Communications.

However, there are few points that need further clarification from the authors prior to publication.

1. The branching ratio for the binary and TBD at 105.7 nm shows a deviation from the general trend which is beyond the uncertainty. It could be due to a different dynamical process happening at that particular wavelength. Can the authors give an explanation for that change?

2. Since the PES and the dynamics play a major role in this photodissociation process. I believe authors should discuss more about that in the main text rather than in the supplementary materials.

3. In Fig 1. D), the experimental spectra and the simulation needs to be labelled.

Reviewer #3 Comments to Author:

This is a most outstanding scientific work! I am extremely impressed by the closure of the circle on the fascinating problem of the VUV photodissociation of the water molecule, the molecule of life as we know it, by Xueming Yang and coworkers.

Besides the great fundamental relevance, are also the wide ranging implications, especially in relation to the very important problem of the source of prebiotic oxygen, that makes this study a landmark study.

As always, the experiments from the Dalian group are truly outstanding. This time we are witnessing a fantastic, very rewarding application of the new powerful VUV free electron laser recently implemented in Dalian that has permitted to unravel the deep VUV photodissociation dynamics of water, showing that the 3-body break-up is very important in the investigated spectral region around 100 nm. Besides being a text-book example in photochemistry, this study clearly shows that oxygen formation by VUV photodissociation of H₂O is likely to be important in the coma of comets, in the interstellar clouds, and even in the Earth's primitive atmosphere, and certainly needs to be included in photochemical models of the interstellar medium.

The manuscript is extremely well written, including the rich and very informative Supplementary Materials. During my long career I have reviewed many exciting papers for Nature and Science, but this one is among the very top, if not the top one!

Congratulations to the authors! This has been a challenging work that has required many skills and people, and deep physical insight. I am very glad that the Dalian group could write what perhaps was the missing chapter on the fascinating story of the photochemistry of the water molecule. Although some further remaining, important surprises may still be around the corner on this problem, with this study a tremendous accomplishment has been made and the scientific community at large should be very thankful to the authors for it.

I have spotted a few minor typos that I believe can be dealt with simply at the editorial level. These are:

1) Abstract: the acronym VUV is introduced two times.

2) page 3, 7 lines from bottom: "...and contribute limited to the O₂ production⁹." Perhaps it should be: "...and contribute **limitedly** to the O₂ production⁹."

3) page 5, Results and Discussion – 3rd line: delete one closing parenthesis.

4) page 9 – Conclusions: 3rd line from bottom of Conclusions: "... incorporated into interstellar photochemical model." It should be plural: "... incorporated into interstellar photochemical **models**."

5) page 19 – Ref. 38: "Yung, Y. L. & Demore, W. B. Photochemistry of Planetary Atmosphere" should be "Yung, Y. L. & **DeMore**, W. B. Photochemistry of Planetary **Atmospheres**".

6) finally, throughout the text there is always (and they are many times) a space between O and its electronic state (3P) or (1D), while in Table 1 and in the Figure captions the spectroscopic notation is always correct (i.e., without interspace). I think it should be corrected throughout.

RECOMMENDATION: Publish immediately!

(Reviewer: Piergiorgio Casavecchia)

Reviewer #4 (Remarks to the Author):

Please find below my referee's report for the manuscript "Three-Body Photodissociation of Water Molecule: An Important Prebiotic Oxygen Source" by Y. Chang et al.. I recommend the paper for publication in Nature Communications as it stands.

The authors present new experimental results showing that three-body dissociation is the dominant pathway following absorption of a photon in the 90 – 107 nm region. The experimentally determined branching ratio for three-body dissociation is combined with a model of the early solar flux and the water molecule absorption cross section to determine that three-body dissociation may result about 20% of the time. This result then argues for including three-body H₂O dissociation as a source for oxygen in the early earth atmosphere.

In addition to the astrophysical applications that rely on a quantitative understanding of the H₂O dissociation paths following VUV excitation, there is intrinsic merit on the basic molecular physics side of the ledger in elucidating these dissociation mechanisms. The paper is well written and comprehensive; the experimental methods are presented at an appropriate level of detail. The experimental techniques, which utilize a pulsed VUV free-electron laser and H-atom tagging, are state of the art. In short, this is a fine piece of experimental work that is well-presented and whose results may impact early-Earth atmospheric modeling.

The paper could use a fairly careful proof-reading as there are numerous awkward grammatical constructions. For example, in the title of the paper "Water Molecule" should be replaced by "the Water Molecule".

I have two relatively small questions for the authors to consider:

1. With the high photon flux of the free-electron laser, is it possible that the water molecule first absorbs a single photon, dissociating into OH + H, and then subsequently the OH radical absorbs a second photon to produce the "three-body" dissociation? Do the experiments (or photon flux plus relevant absorption cross sections) rule out this possibility?
2. In Figure 4, the solar flux is binned in large wavelength increments (perhaps 2 nm?). But the solar spectrum is highly structured in the region of interest. For example the solar flux at Lyman alpha completely dominates the spectrum in the 120 nm region. It's not entirely clear why such crude binning is used for the solar flux, nor what the consequences might be (for the estimated 20% three-body value) of using a more appropriate solar spectrum.

It would be helpful if the authors could clarify these two points.

Responses to reviewers' comments

Reviewer #1

The manuscript submitted by Chang et al. represents an outstanding contribution in an important problem regarding the provenance of molecular oxygen on Earth and other planets by a detailed study of the three-body photodissociation of water by using tunable VUV free electron laser radiation in combination with the high resolution hydrogen-atom Rydberg tagging time-of-flight technique. These unique experimental capabilities have allowed the determination of quantum yields for the two possible three-body photodissociation channels at different excitation wavelengths in the range 92-109 nm, which have permitted to evaluate the role of VUV three-body photodissociation of water in the production of prebiotic molecular oxygen in Earth's primitive atmosphere. These results will be very important for the interstellar photochemical models. The paper is very good written, the results are clearly presented with the help of the supplementary material and the conclusions are supported by the data. I recommend publication of this work as it is.

Author Reply: Thank you very much for your comments !

Reviewer #2

The manuscript reports a new pathway for oxygen production by three body dissociation of H₂O molecules with VUV radiation utilizing the newly developed free electron laser (FEL) at the Dalian Coherent Light Source (DCLS). This is the first comprehensive quantitative observation of TBD of water molecules. I believe this study is greatly important for the better understanding of different oxygen sources in the atmosphere therefore, beneficial for future investigations. Since O atom is a precursor for many atmospheric molecules this will also lay a ground work for studying new photochemical processes. Therefore, I strongly recommend the article for the publication of Nature Communications.

However, there are few points that need further clarification from the authors prior to publication.

1. The branching ratio for the binary and TBD at 105.7 nm shows a deviation from the general trend which is beyond the uncertainty. It could be due to a different dynamical process happening at that particular wavelength. Can the authors give an explanation for that change?

Author Reply: Thank you very much for your comments! Indeed, the branching ratio for the binary and TBD at 105.7 nm shows a deviation from the general trend, i.e., the binary channel from 105.7 nm photolysis is larger than that from its neighboring wavelengths. The TKER spectrum at 105.7 nm shown in Supplementary Figure 3C also displays clear sharp features in the middle translational energy range, which are ascribed to high rotational levels of OH ($X, v=0$) radicals, suggesting a little different dynamical process. Such dynamical source is not immediately clear, however, since the potential energy surfaces of these Rydberg states of water are lacking. As reported by Fillion et al. (Ref. 28), the intense features of the absorption spectrum (98-114 nm), are dominated by the $nd\leftarrow 1b_1$ transition series. At 105.7 nm, the water molecule was excited to $4d$ Rydberg state with 1B_2 symmetry. A bent-linear interaction between this 1B_2 state and the $^1B_2 3pb_2$ state (the state of which has a quasilinear geometry and arises from the excitation of the $3a_1$ orbit) may occur. Thus, the water molecule can undergo a fast conversion from the initial excited Rydberg state to the 1B_2 state via an avoid crossing between them in the region of 120° (Ref. 28), and then predissociate to the dissociative $^1A_2 3pb_2$ state due to the Renner-Teller coupling between the 1B_2 and 1A_2 states. The molecules on the 1A_2 state can further couple to the \tilde{A} state yielding H+OH (X) products. While at other photolysis wavelengths, the water molecules mainly undergo a fast dissociation on the \tilde{B}^1A_1 state surface after

multi-step internal conversions from the initial excited Rydberg state to the \tilde{B} state. We have added this part in the Supplementary Note 3.

2. Since the PES and the dynamics play a major role in this photodissociation process. I believe authors should discuss more about that in the main text rather than in the supplementary materials.

Author Reply: Thank you very much for your suggestion! We have added one paragraph in the main text to discuss more about the dissociation process. See Page 7-8 in the main text.

3. In Fig 1. D), the experimental spectra and the simulation needs to be labelled.

Author Reply: Thank you very much for your suggestion! We have added the labels in Fig 1.D).

Reviewer #3

This is a most outstanding scientific work! I am extremely impressed by the closure of the circle on the fascinating problem of the VUV photodissociation of the water molecule, the molecule of life as we know it, by Xueming Yang and coworkers. Besides the great fundamental relevance, are also the wide ranging implications, especially in relation to the very important problem of the source of prebiotic oxygen, that makes this study a landmark study. As always, the experiments from the Dalian group are truly outstanding. This time we are witnessing a fantastic, very rewarding application of the new powerful VUV free electron laser recently implemented in Dalian that has permitted to unravel the deep VUV photodissociation dynamics of water, showing that the 3-body break-up is very important in the investigated spectral region around 100 nm. Besides being a text-book example in photochemistry, this study clearly shows that oxygen formation by VUV photodissociation of H₂O is likely to be important in the coma of comets, in the interstellar clouds, and even in the Earth's primitive atmosphere, and certainly needs to be included in photochemical models of the interstellar medium. The manuscript is extremely well written, including the rich and very informative Supplementary Materials. During my long career I have reviewed many exciting papers for Nature and Science, but this one is among the very top, if not the top one!

Congratulations to the authors! This has been a challenging work that has required many skills and people, and deep physical insight. I am very glad that the Dalian group could write what perhaps was the missing chapter on the fascinating story of the photochemistry of the water molecule. Although some further remaining, important surprises may still be around the corner on this problem, with this study a tremendous accomplishment has been made and the scientific community at large should be very thankful to the authors for it.

I have spotted a few minor typos that I believe can be dealt with simply at the editorial level. These are:

1) Abstract: the acronym VUV is introduced two times.

2) page 3, 7 lines from bottom: "...and contribute limited to the O₂ production⁹." Perhaps it should be: "...and contribute **limitedly** to the O₂ production⁹."

3) page 5, Results and Discussion – 3rd line: delete one closing parenthesis.

4) page 9 – Conclusions: 3rd line from bottom of Conclusions: "... incorporated into interstellar photochemical model." It should be plural: "... incorporated into interstellar photochemical **models**."

5) page 19 – Ref. 38: "Yung, Y. L. & Demore, W. B. Photochemistry of Planetary Atmosphere" should be

"Yung, Y. L. & **DeMore**, W. B. Photochemistry of Planetary **Atmospheres**".

6) finally, throughout the text there is always (and they are many times) a space between O and its electronic state (3P) or (1D), while in Table 1 and in the Figure captions the spectroscopic notation is always correct (i.e., without interspace). I think it should be corrected throughout.

RECOMMENDATION: Publish immediately!

Author Reply: Thank you very much for your comments! We have revised the typos in the paper.

Reviewer #4

Please find below my referee's report for the manuscript "Three-Body Photodissociation of Water Molecule: An Important Prebiotic Oxygen Source" by Y. Chang et al.. I recommend the paper for publication in Nature Communications as it stands.

The authors present new experimental results showing that three-body dissociation is the dominant pathway following absorption of a photon in the 90 – 107 nm region. The experimentally determined branching ratio for three-body dissociation is combined with a model of the early solar flux and the water molecule absorption cross section to determine that three-body dissociation may result about 20% of the time. This result then argues for including three-body H₂O dissociation as a source for oxygen in the early earth atmosphere.

In addition to the astrophysical applications that rely on a quantitative understanding of the H₂O dissociation paths following VUV excitation, there is intrinsic merit on the basic molecular physics side of the ledger in elucidating these dissociation mechanisms. The paper is well written and comprehensive; the experimental methods are presented at an appropriate level of detail. The experimental techniques, which utilize a pulsed VUV free-electron laser and H-atom tagging, are state of the art. In short, this is a fine piece of experimental work that is well-presented and whose results may impact early-Earth atmospheric modeling.

The paper could use a fairly careful proof-reading as there are numerous awkward grammatical constructions. For example, in the title of the paper "Water Molecule" should be replaced by "the Water Molecule".

I have two relatively small questions for the authors to consider:

1. With the high photon flux of the free-electron laser, is it possible that the water molecule first absorbs a single photon, dissociating into OH + H, and then subsequently the OH radical absorbs a second photon to produce the "three-body" dissociation? Do the experiments (or photon flux plus relevant absorption cross sections) rule out this possibility?

Author Reply: Thank you very much for your comments! In our experiments, we have reduced the FEL laser power down as low as possible (<10 μj /pulse), and kept the laser spot unfocusing (1-2 mm diameter). This leads to very low possibility to absorb a second photon to dissociate the OH radicals. Actually, the energetic OH radicals absorb another VUV photon will produce H atoms with the much higher velocity. We have not observed such H atom signals in the early arriving time of the TOF spectra, thus we have ruled out the secondary dissociation.

2. In Figure 4, the solar flux is binned in large wavelength increments (perhaps 2 nm?). But the solar spectrum is highly structured in the region of interest. For example the solar flux at Lyman alpha completely dominates the spectrum in the 120 nm region. It's not entirely clear why such crude binning is used for the solar flux, nor what the consequences might be (for the estimated 20% three-body value) of using a more appropriate solar spectrum.

Author Reply: Thank you very much for your comments! Indeed, numerous references reported the modern sun spectrum with high resolution (*Astrophys. J.* **761**, 166 (2012); *Astrophys. J.*, **757**, :95 (2012)), but the sun spectrum at early stages (~4 billion years ago) all came from the modelling. In this work the sun spectrum at the earliest stage (~10 million years) was adopted from Subrata et al. (*PNAS*, 110, 17650 (2013)) and Zahnle et al. (*Rev. Geophys. Space Phys.* 20, 280 (1982)), the only spectrum at this stage we can find in the references. The difference between the modern sun spectrum and that at early stages is that the VUV radiation emitted from the nascent sun is about 10 times stronger than it does today, which makes the water photochemistry more important. The other VUV solar flux also can be used, which may modify the yield of O-production a little (still around 20%

quantum yield), but the final conclusion holds. We have prepared the original data in the Supplementary Material that allow readers to convolute the existing H₂O absorption data and the present branching data with any user-chosen, λ -dependent Solar flux.

The reviewer identifies some typos. We have checked the typos in the paper.

We very much hope that you judge these responses to be appropriate and hope to see the manuscript published in a future issue of *Nature Communications*.

Yours sincerely,

K. J. Yuan

REVIEWERS' COMMENTS

Reviewer #2 (Remarks to the Author):

The current version addresses all my questions. In my opinions I recommend it to be considered for publication.

Reviewer #4 (Remarks to the Author):

The authors have thoroughly addressed the two questions that I raised. I recommend publication without further review.

Reviewer #2 (Remarks to the Author):

The current version addresses all my questions. In my opinions I recommend it to be considered for publication.

Reviewer #4 (Remarks to the Author):

The authors have thoroughly addressed the two questions that I raised. I recommend publication without further review.

Author reply: Thank you very much for your comments. We are very glad to hear that the reviewers 2 and 4 find our manuscript publishable.